# Effects of Water Potential on Anaerobic Methane Production and a Microbial Consortium

Jin Yeo [1,2], Chang-Gyu Kim [1], Jun-Heong Lee [1], Eunhye Song [3] and Young-Man Yoon [1,2,*]

[1] Department of Plant Life & Environmental Science, Hankyong National University, Anseong 17579, Gyeonggi, Republic of Korea
[2] Biogas Research Center, Hankyong National University, Anseong 17579, Gyeonggi, Republic of Korea
[3] Plant Engineering Center, Institute for Advanced Engineering, Yongin 17528, Gyeonggi, Republic of Korea
[*] Correspondence: yyman@hknu.ac.kr; Tel.: +82-31-670-5086

**Abstract:** This study probed the effect of the water potential ($\Psi$) on anaerobic methane production and a microbial consortium. The $\Psi$ level of the investigated anaerobic digester (n = 20) was in the range from $-0.10$ to $-2.09$ MPa with a mean value of $-1.23$ MPa, and the $\Psi$ level of the anaerobic digester was significantly correlated with the SCOD, TKN, $NH^{4+}$-N, alkalinity, salinity ($S_{PS}$), $NH_4^+$, $Na^+$, $K^+$, $Cl^-$, $NO_3^-$, and $PO_4^{3-}$ ($p < 0.001$). The maximum methane production rate ($R_m$) of the Control ($-0.40$ MPa) was 8.11 mL day$^{-1}$ and decreased to 1.70 mL day$^{-1}$ at $-3.91$ MPa (K5), and the lag growth phase time ($\lambda$) was delayed to 35.96 and 25.34 days at $-2.85$ MPa (K4) and $-3.91$ MPa (K5), respectively. The ultimate methane potential ($B_u$) was 0.264 Nm$^3$ kg$^{-1}$-VS$_{added}$ for the Control, and when $\Psi$ was adjusted, $B_u$ increased to 0.278 Nm$^3$ kg$^{-1}$-VS$_{added}$ at $-1.49$ MPa (K3) but decreased to 0.203 and 0.172 Nm$^3$ kg$^{-1}$-VS$_{added}$ at $-2.85$ MPa (K4) and $-3.91$ MPa (K5), respectively. Therefore, the methane yield was inhibited due to the decrease in $\Psi$, and the methane yield is predicted to be inhibited from about $-1.65$ MPa. In the genus-level taxonomic classification of the microbial community, the relative abundance of *Methanosarcina* decreased significantly to 36.76% at $-3.91$ MPa (K5) compared to 58.15% for the Control; however, the relative abundance of *Methanoculleus* significantly increased to 35.16% at $-3.91$ MPa (K5) compared to 14.85% for the Control.

**Keywords:** water potential; osmotic potential; salt stress; anaerobic digestion; methanogens; methane yield

## 1. Introduction

Anaerobic digestion technology is a biological conversion technology to produce methane ($CH_4$) from organic matter using a sequential microbial reaction. The operating factors of a typical anaerobic digestion process are those factors known to affect the activity of methanogens, such as the pH, alkalinity, temperature, total volatile fatty acids (TVFAs), and hydraulic retention time (HRT) [1–3]. In order to maximize the efficiency of methane production of an anaerobic digester, it is important to maintain the conditions such that they optimally support the activity of the methanogens, which is inhibited outside of the optimal range of operating factors of the digester. Kroeker et al. [4] classified the factors that inhibit the activity of methanogenic bacteria in an anaerobic digestion process into two types: toxicity and inhibition. Toxicity is represented by the complete cessation of methanogenic activity, and inhibition occurs as a result of reducing the rate and extent of methanogenesis. In particular, ammonia ($NH_3$), hydrogen sulfide ($H_2S$), cations ($Na^+$, $K^+$, $Ca^{2+}$, $Mg^{2+}$), long chain fatty acids (LCFA), heavy metals, etc., have been reported as inhibitors of the activity of methanogens [5].

In general, organic waste from resources such as livestock manure, food waste, and sewage sludge is used in the anaerobic digestion process [6,7]. In Korea, since the methane production efficiency is low in the case of anaerobic digestion of livestock manure and

sewage sludge, a co-digestion method, in which these two types of waste are mixed with food waste with relatively high methane production efficiency, is used [8]. However, the food waste generated in Korea contains a high concentration of salt of approximately 3% (based on dry matter content) [9], which causes an operational problem for the anaerobic digestion of food waste owing to salt stress. The inhibitory effect of salt on anaerobic digestion is known to be due to cations such as sodium ion ($Na^+$), potassium ion ($K^+$), calcium ion ($Ca^{2+}$), and magnesium ion ($Mg^{2+}$) [10]. In previous studies related to cation inhibition, the inhibition of anaerobic digestion efficiency was reported to occur in the concentration range of $Na^+$ from 4100 to 11,000 mg $L^{-1}$, $K^+$ from 3000 to 28,000 mg $L^{-1}$, $Ca^{2+}$ from 5000 to 8000 mg $L^{-1}$, and $Mg^{2+}$ from 729 to 3000 mg $L^{-1}$ [9,11–18]. As such, the cation inhibitory effect on anaerobic digestion is diverse in that it occurs across a wide concentration range of each ionic species, making it difficult to manage the inhibitory concentration for each type of cation when operating an anaerobic digester. Therefore, commercial anaerobic digestion reactors are operated without taking into consideration the inhibition of the anaerobic digester by the presence of excess cations.

The water potential ($\Psi$) is an indicator of the state of the potential energy of water and can be used to evaluate the availability of water for microorganisms. In an anaerobic digester, the $\Psi$ is mainly composed of the osmotic potential ($\Psi_o$), which represents the colligative property of the solution affected by the concentration of salt, organic acid, etc. In general, high concentrations of salt are known to lower $\Psi$ and induce osmotic stress by cell dehydration, which results in the inhibition of microbial activity and metabolism [19–23]. Generally, the cytoplasm of microorganisms has a relatively low $\Psi$ due to the concentration of various solutes [23,24], but severe osmotic stress induces plasmolysis [25,26] and cytolysis [27] of microorganisms. Excessive salt in the anaerobic digester may therefore severely inhibit the activity and metabolism of anaerobic microorganisms because of the low $\Psi$. However, studies on the effect of salt on the anaerobic digestion efficiency have revealed a range of inhibitory concentrations for each ionic species, yet the dependence of the anaerobic digestion efficiency on $\Psi$, which represents a comprehensive indicator of salt stress, has not been reported. This motivated our study, in which we attempted to understand the effect of $\Psi$ on anaerobic methane production by the microbial reaction of methanogens. The $\Psi$ level of 20 anaerobic digesters was evaluated, and the effect of the physicochemical factors of these anaerobic digesters on the $\Psi$ was analyzed. In addition, a batch-type anaerobic reactor was operated by adjusting the $\Psi$ using potassium chloride (KCl). This enabled us to analyze the changes in the characteristics of a community of anaerobic microorganisms according to the changes in $\Psi$ using next-generation sequencing (NGS) technology.

## 2. Materials and Methods

### 2.1. Materials

The physicochemical properties and the value of $\Psi$ were determined for anaerobic digestates collected from 20 anaerobic digestion facilities in operation in Korea (Table 1). The facilities we selected for our investigation were those in which livestock manure, sewage sludge, and food waste are anaerobically digested or co-digested, and these facilities were operated under mesophilic conditions.

**Table 1.** Operating status of the facilities investigated in this study.

| No. | Input Materials | Capacity (m³ Day⁻¹) | Digester Temperature (°C) | HRT [1] (Days) | Location |
|---|---|---|---|---|---|
| 1 | Sewage sludge | 2863 | 37.5 | 15.5 | Gangnam, Seoul |
| 2 | Pig slurry, food wastewater | 200 | 36.8 | 25.0 | Goyang, Gyeonggi |
| 3 | Pig slurry, food wastewater, livestock byproducts | 150 | 36.0 | 30.0 | Nonsan, Chungnam |
| 4 | Sewage sludge, food wastewater | 2000 | 37.5 | 25.0 | Dongnae, Busan |
| 5 | Pig slurry, food waste, sewage sludge | 320 | 37.0 | 35.0 | Seosan, Chungnam |
| 6 | Sewage sludge | 2700 | 37.5 | 30.0 | Seongnam, Gyeonggi |
| 7 | Sewage sludge, food wastewater | 140 | 34.5 | 20.0 | Ansan, Gyeonggi |
| 8 | Pig slurry, food wastewater, sewage sludge | 384 | 36.0 | 15.0 | Anyang, Gyeonggi |
| 9 | Pig slurry, food waste | 130 | 36.0 | 20.0 | Yangsan, Gyeongnam |
| 10 | Pig slurry, food wastewater | 100 | 40.0 | 35.0 | Yeoncheon, Gyeonggi |
| 11 | Food waste, slaughterhouse byproducts | 220 | 40.0 | 55.0 | Wonju, Gangwon |
| 12 | Pig slurry, food wastewater | 100 | 38.0 | 40.0 | Icheon, Gyeonggi |
| 13 | Food wastewater | 650 | 39.0 | 43.0 | Seo, Incheon |
| 14 | Food wastewater | 300 | 36.0 | 25.0 | Jeonju, Jeonbuk |
| 15 | Pig slurry, food wastewater | 125 | 38.0 | 11.0 | Seogwipo, Jeju |
| 16 | Food wastewater | 150 | 39.5 | 26.0 | Cheongju, Chungbuk |
| 17 | Sewage sludge, food wastewater | 525 | 38.0 | 30.7 | Chuncheon, Gangwon |
| 18 | Food waste | 80 | 38.0 | 19.0 | Chungju, Chungbuk |
| 19 | Pig slurry, food wastewater | 100 | 37.0 | 35.0 | Hongcheon, Gangwon |
| 20 | Food wastewater | 120 | 39.5 | 30.0 | Hwaseong, Gyeonggi |

[1] Hydraulic retention time.

*2.2. Water Potential (Ψ) Analysis*

Ψ is defined as the difference in the chemical potential of water (J mol⁻¹) per unit volume (J m⁻³) between a given water sample and pure free water at the same temperature. This is expressed as Equation (1). Rearrangement of the units in Equation 1 shows that Ψ is expressed in terms of the pressure (MPa), which is frequently used as the unit in Ψ measurements [28].

$$\Psi = \frac{\mu_W - \mu^{\circ}{}_W}{V_W} \tag{1}$$

where Ψ is the water potential (MPa), $\mu_W$ is the chemical potential of the water, $\mu^{\circ}{}_W$ is the chemical potential of the free water at the same temperature, and $V_W$ is the partial molar volume of water in the system. The Ψ was measured using a dew point hygrometer (WP4C, METER Group, Inc., Pullman, MA, USA) based on the principle of the chilled mirror dew point. When the air in the chamber containing the sample flows over the cooling mirror, dew forms on the surface of the mirror. The Ψ of a liquid sample is found by relating the sample Ψ reading to the saturation vapor pressure of air in equilibrium with the sample. The relationship between the sample Ψ and the saturation vapor pressure of the air is determined using Equation (2) [29].

$$\Psi = \frac{RT}{M_w} ln \frac{e_s(T_d)}{e_s(T_s)} \tag{2}$$

where Ψ is the water potential (MPa), $e_s(T_d)$ is the saturation vapor pressure of the air at dew point temperature, $e_s(T_s)$ is the saturation vapor pressure at sample temperature, $R$ is the gas constant, 8.31 J mol⁻¹ K⁻¹, $T$ is the temperature of the sample in Kelvin, and $M_w$ is the molecular mass of water.

### 2.3. Methane Production Potential

2.3.1. Theoretical Methane Potential ($B_{th}$)

The theoretical methane potential ($B_{th}$) was calculated stoichiometrically using Boyle's equation based on the elemental analysis results of the samples (Equations (3) and (4)).

$$C_aH_bO_cN_dS_e + \left(a - \frac{b}{4} - \frac{c}{2} + \frac{3d}{4} + \frac{e}{2}\right)H_2O \rightarrow \left(\frac{a}{2} + \frac{b}{8} - \frac{c}{4} - \frac{3d}{8} - \frac{e}{4}\right)CH_4 + \left(\frac{a}{2} - \frac{b}{8} + \frac{c}{4} + \frac{3d}{8} + \frac{e}{4}\right)CO_2 + dNH_3 + eH_2S \quad (3)$$

$$B_{th}\left(Nm^3\ kg^{-1} - VS_{added}\right) = 22.4 \times \left[\frac{(4a + b - 2c - 3d - 2e)/8}{12a + b + 16c + 14d + 32e}\right] \quad (4)$$

2.3.2. Water Potential ($\Psi$) Inhibition Assay

The ultimate methane potential ($B_u$) was assessed by the biochemical methane potential assay [30]. To assess the $B_u$ in the $\Psi$ inhibition assay, a batch-type anaerobic reactor was operated under mesophilic conditions (38 °C). The anaerobic inoculum was collected from a farm-scale anaerobic digester located in Icheon City, Republic of Korea. The chemical properties of the inoculum are provided in Table 2.

**Table 2.** Chemical composition of inoculum.

| Parameters | pH | TS [1] | VS [2] | TCOD [3] | SCOD [4] | TKN [5] | NH$_4^+$-N [6] | Alkalinity | TVFAs [7] |
|---|---|---|---|---|---|---|---|---|---|
| | (-) | (mg L$^{-1}$) | (mg L$^{-1}$) | (mg L$^{-1}$) | (mg L$^{-1}$) | (mg L$^{-1}$) | (mg L$^{-1}$) | (mg L$^{-1}$ as CaCO$_3$) | (mg L$^{-1}$ as Acetate) |
| Inoculum | 7.96 | 30,044 | 14,767 | 21,050 | 6012 | 2201 | 1314 | 6389 | 113 |

[1] Total solid, [2] Volatile solid, [3] Total chemical oxygen demand, [4] Soluble chemical oxygen demand, [5] Total Kjeldahl nitrogen, [6] Ammonium nitrogen, [7] Total volatile fatty acids.

The batch anaerobic reactors for the $\Psi$ inhibition assay were prepared as in Table 3. The inoculum for the $\Psi$ inhibition assay was kept under mesophilic conditions at 38 °C for two weeks to remove any remaining biodegradable fraction. The substrate to inoculum ratio in all anaerobic batch reactors was equal to 0.5 (g-VS$_{substrate}$/g-VS$_{inoculum}$). The $\Psi$ of each batch anaerobic reactor was adjusted to −0.57, −0.89, −1.49, −2.85, and −3.91 MPa using the KCl solution for the $\Psi$ inhibition assay. The working volume for anaerobic batch fermentation was 200 mL of a 250 mL serum bottle. The headspace of the serum bottle was filled with N$_2$ gas and sealed with a butyl rubber stopper. The anaerobic batch reactors for each sample and blank were incubated for up to 78 days in the convection incubator and manually mixed each day during the fermentation period. Then, the anaerobic batch reactions for each sample and blank were performed in triplicate. The biochemical methane potential was calculated based on the volatile solid (VS) content. The biochemical methane potentials of the samples were corrected using the blank value and calibrated under standard temperature and pressure (STP) conditions (0 °C, 1 atm). The modified Gompertz model (Equation (5)) was employed to interpret the progress of cumulative methane production. This enabled the cumulative methane production data to be optimized using the following equations [31].

$$M = P \times \exp\left\{-exp\left[\frac{R_m \cdot e}{P}(\lambda - t) + 1\right]\right\} \quad (5)$$

where $M$ is the cumulative methane production (mL), $t$ is the anaerobic fermentation time (days), $P$ is the final methane production (mL), $e$ is the exp (1), $R_m$ is the maximum methane production rate (mL day$^{-1}$), and $\lambda$ represents the lag growth phase time (days). The cumulative methane production curves in the $\Psi$ inhibition assay were optimized with SigmaPlot (SigmaPlot Version 12.5, Systat Software Inc., Chicago, IL, USA) using the modified Gompertz model. The inhibition degree in the $\Psi$ inhibition assay was used to describe the adverse impact of water stress on the methane yield by Equation (6). Herein,

$ID$ is the inhibition degree (%), $B_{u-C}$ is the ultimate methane potential (Nm$^3$ kg$^{-1}$-VS$_{added}$) in the Control, and $B_{u-T}$ is the ultimate methane potential (Nm$^3$ kg$^{-1}$-VS$_{added}$) after the treatment.

$$ID(\%) = \left(1 - \frac{B_{u-C}}{B_{u-T}}\right) \tag{6}$$

**Table 3.** Experimental conditions for the water potential ($\Psi$) inhibition assay.

| Parameters | | Blank | Control | Treatments | | | | |
| --- | --- | --- | --- | --- | --- | --- | --- | --- |
| | | | | K1 | K2 | K3 | K4 | K5 |
| Inoculum (mL) | | 70 | 70 | 70 | 70 | 70 | 70 | 70 |
| Substrate [1] (g) | | 0 | 0.52 | 0.52 | 0.52 | 0.52 | 0.52 | 0.52 |
| I/S ratio [2] (-) | | - | 0.5 | 0.5 | 0.5 | 0.5 | 0.5 | 0.5 |
| WP [3] adjusting solution | KCl conc. (%) | - | - | 0.46 | 0.98 | 1.95 | 4.23 | 5.85 |
| | Volume (mL) | - | - | 130 | 130 | 130 | 130 | 130 |
| DW [4] volume (mL) | | 130 | 130 | - | - | - | - | - |
| Operation volume (mL) | | 200 | 200 | 200 | 200 | 200 | 200 | 200 |
| $\Psi$ [5] (MPa) | | −0.36 (0.03) [6] | −0.40 (0.03) | −0.57 (0.03) | −0.89 (0.03) | −1.49 (0.04) | −2.85 (0.01) | −3.91 (0.02) |
| K$^+$ conc. [7] (%) | | N.A. [8] | N.A. | 0.24 | 0.51 | 1.02 | 2.22 | 3.07 |

[1] D-Glucose used as substrate, [2] Substrate to inoculum ratio (g-VS$_{substrate}$/g-VS$_{inoculum}$), [3] Water potential, [4] Distilled water. [5] Water potential of batch anaerobic reactor, [6] Standard deviation, [7] K$^+$ concentration, [8] Not analyzed.

### 2.4. Analysis of the Microbial Consortium

DNA was extracted using a DNA kit (GD141-050 Gram positive, Biofact, Seoul, Republic of Korea) according to the manufacturer's instructions. The extracted DNA was quantified using Quant-IT PicoGreen (Invitrogen, Waltham, MA, USA).

DNA Extraction and Quantification

The sequencing libraries were prepared according to the Illumina 16S Metagenomic Sequencing Library protocols to amplify the V3 and V4 regions. The input gDNA 2 ng was PCR amplified with 5× reaction buffer, 1 mM of dNTP mix, 500 nM each of the universal F/R PCR primer, and Herculase II fusion DNA polymerase (Agilent Technologies, Santa Clara, CA, USA). The cycling conditions for the 1st PCR were 3 min at 95 °C for heat activation, and 25 cycles of 30 s at 95 °C, 30 s at 55 °C, and 30 s at 72 °C, followed by a 5 min final extension at 72 °C. The universal primer pair with Illumina adapter overhang sequences used for the first amplifications were as follows: V3-F: 5′-TCGTCGGCAGCGTCAGATGTGTATAAGAGACAGCCTACGGGNGGCWGCAG-3′, V4-R: 5′-GTCTCGTGGGCTCGGAGATGTGTATAAGAGACAGGACTACHVGGGTATCTAATCC-3′. The 1st PCR product was purified with AMPure beads (Agencourt Bioscience, Beverly, MA, USA). Following purification, 2 μL of the 1st PCR product was PCR amplified for final library construction containing the index using the NexteraXT Indexed Primer. The cycling conditions for the 2nd PCR were the same as for the 1st PCR except that 10 cycles were used. The PCR product was purified with AMPure beads. The final purified product was then quantified using qPCR according to the qPCR Quantification Protocol Guide (KAPA Library Quantification kits for Illumina Sequencing platforms) and qualified using the TapeStation D1000 ScreenTape (Agilent Technologies, Waldbronn, Germany). The paired-end (2 × 300 bp) sequencing was performed by the Macrogen online sequencing system using the MiSeq™ platform (Illumina, San Diego, CA, USA).

*2.5. Chemical Analysis*

The physicochemical properties of the collected anaerobic digestates were determined and the contents of the reactors in the $\Psi$ inhibition assay were analyzed based on standard methods [29]. The total solid (TS), volatile solid (VS), fixed solid (FS), total suspended solid (TSS), volatile suspended solid (VSS), fixed suspended solid (FSS), total chemical oxygen demand (TCOD), soluble chemical oxygen demand (SCOD), total Kjeldahl nitrogen (TKN), ammonium nitrogen ($NH_4^+$-N), and total volatile fatty acids (TVFAs) were analyzed to determine the organic composition. In addition, the pH, alkalinity, salinity ($S_{PS}$), cations ($Na^+$, $K^+$, $Mg^{2+}$, $Ca^{2+}$), and anions ($NO_3^-$, $Cl^-$, $SO_4^{2-}$, $PO_4^{3-}$) were analyzed to determine the chemical properties. More specifically, the VFAs were measured using a gas chromatograph (GC2010, Shimadzu Scientific Instruments, Inc., Columbia, MD, USA) equipped with a flame ionization detector with an automatic sampler. The concentrations of the soluble cations ($Na^+$, $K^+$, $Mg^{2+}$, $Ca^{2+}$) were determined by analyzing the filtrate from which TSS was removed using inductively coupled plasma optical emission spectrometry (ICP-OES) (Avio 550 Max, PerkinElmer, USA). The concentrations of the soluble anions ($NO_3^-$, $Cl^-$, $SO_4^{2-}$, $PO_4^{3-}$) in the filtrate from which TSS was removed were analyzed by ion chromatography (IC) (833 Basic IC plus, Metrohm, Switzerland). The salinity was defined as the practical salinity ($S_{PS}$), which was calculated from the electrical conductivity measured by an electrical conductivity meter (CM-42X, TOADKK, Japan) [32]. In the $\Psi$ inhibition assay, the total gas production was measured daily for the first seven days and then every two or three days. The gas that was produced displaced an acidified brine solution in a burette and the volume of the displaced solution was recorded after correcting for atmospheric pressure [33]. The $CH_4$ and $CO_2$ concentrations in the gas samples were determined using a gas chromatograph (Clarus 680, PerkinElmer, Inc., Waltham, MA, USA) equipped with a thermal conductivity detector and a HayeSepQ packed column (CRS, Inc., Louisville, KY, USA). The column was operated with helium as a carrier gas at a flow rate of 5 mL/min. The temperatures of the injector, oven, and detector were set to 150, 90, and 150 °C, respectively [34]. Total ammonium nitrogen (TAN) has two forms, $NH_4^+$-N and $NH_3$-N, which were calculated according to Equation (7) [35], where $T$ is the absolute temperature (K).

$$Free\ NH_3 - N = TAN \times \left[1 + \frac{10^{-pH}}{10^{-(0.09018 + \frac{2729.92}{T})}}\right]^{-1} \tag{7}$$

*2.6. Statistical Analysis*

In this study, to examine the physicochemical parameters that affect the $\Psi$ of the anaerobic digester, the Pearson correlation coefficients between $\Psi$ and the physicochemical parameters (pH, WC, TS, VS, FS, TSS, VSS, FSS, TCOD, SCOD, TKN, $NH_4^+$-N, alkalinity, TVFAs, $S_{PS}$, $Na^+$, $K^+$, $Mg^{2+}$, $Ca^{2+}$, $NO_3^-$, $Cl^-$, $SO_4^{2-}$, $PO_4^{3-}$) were analyzed by R studio (R Version 4.1.3, R Foundation for Statistical Computing, Vienna, Austria). The tables in this article present the mean values and standard deviations of the data obtained from the experiments. The statistical analysis of the results of this experiment was conducted using the general linear model (GLM) procedure of the SAS program package (SAS ver. 9.4, SAS Instrument Inc., Cary, NC, USA), and the significant difference ($p < 0.05$) of the mean between treatments was tested using Duncan's multiple range test.

## 3. Results and Discussion

*3.1. Correlation Factors with $\Psi$*

The physicochemical properties of 20 anaerobic digesters operating in Korea were investigated, and the results are reported in Tables 4 and 5. The pH of the anaerobic digester had mean, maximum, and minimum values of 7.86, 8.78, and 6.89, respectively. In some facilities, it was necessary to adjust the pH of the digester to lower the pH, but the mean value was within the normal operating range. In addition, the TKN (4.91 g $L^{-1}$) and alkalinity (16.82 g $L^{-1}$) were in the optimum operating range of an anaerobic digester.

The $\Psi$ of the anaerobic digester had mean, maximum, and minimum values of $-1.23$ MPa, $-0.10$ MPa, and $-2.09$ MPa, respectively.

**Table 4.** Physicochemical properties of anaerobic digestates (n = 20) investigated in this study.

| Parameters | | Mean | Median | Maximum | Minimum | r [15] |
|---|---|---|---|---|---|---|
| $\Psi$ [1] (MPa) | | $-1.23$ | $-1.25$ | $-0.10$ | $-2.09$ | - |
| pH (-) | | 7.86 | 7.83 | 8.78 | 6.89 | $-0.50$ ** |
| WC [2] (wt. %, w.b.) | | 96.20 | 96.30 | 98.54 | 92.13 | 0.42 * |
| TCOD [3] (g L$^{-1}$) | | 30.95 | 32.34 | 52.43 | 14.30 | $-0.45$ * |
| SCOD [4] (g L$^{-1}$) | | 9.33 | 9.67 | 24.63 | 0.00 | $-0.68$ *** |
| TKN [5] (g L$^{-1}$) | | 4.91 | 5.66 | 6.95 | 1.05 | $-0.87$ *** |
| TAN [6] (g L$^{-1}$) | | 3.27 | 3.30 | 5.19 | 0.30 | $-0.91$ *** |
| NH$_4^+$-N [7] (g L$^{-1}$) | | 2.87 | 2.87 | 4.57 | 0.30 | - |
| Alkalinity (g L$^{-1}$ as CaCO$_3$) | | 16.82 | 16.57 | 41.60 | 1.45 | $-0.61$ *** |
| TVFAs [8] | (cmol$_c$ L$^{-1}$ as acetate) | 1.30 | 0.50 | 10.34 | 0.19 | $-0.12$ |
| | (mg L$^{-1}$ as acetate) | 778 | 302 | 6210 | 113 | - |
| Solid content | TS [9] (g L$^{-1}$) | 37.96 | 37.03 | 78.72 | 14.60 | $-0.42$ * |
| | VS [10] (g L$^{-1}$) | 20.00 | 20.15 | 28.71 | 10.62 | $-0.50$ ** |
| | FS [11] (g L$^{-1}$) | 17.96 | 16.78 | 53.63 | 3.98 | $-0.32$ |
| | TSS [12] (g L$^{-1}$) | 31.04 | 27.85 | 64.90 | 15.43 | $-0.06$ |
| | VSS [13] (g L$^{-1}$) | 20.51 | 19.85 | 33.80 | 9.63 | $-0.28$ |
| | FSS [14] (g L$^{-1}$) | 15.18 | 10.53 | 38.65 | 3.95 | $-0.33$ |

[1] Water potential, [2] Water content, [3] Total chemical oxygen demand, [4] Soluble chemical oxygen demand, [5] Total Kjeldahl nitrogen, [6] Total ammonium nitrogen (NH$_4^+$-N + NH$_3$-N), [7] Ammonium nitrogen, [8] Total volatile fatty acids, [9] Total solid, [10] Volatile solid, [11] Fixed solid, [12] Total suspended solid, [13] Volatile suspended solid, [14] Fixed suspended solid, [15] Pearson correlation coefficient between the water potential and other parameters (* $p < 0.05$, ** $p < 0.01$, *** $p < 0.001$).

**Table 5.** Chemical properties of anaerobic digestates (n = 20) investigated in this study.

| Parameters | | Mean | Median | Maximum | Minimum | r [3] |
|---|---|---|---|---|---|---|
| $\Psi$ [1] (MPa) | | $-1.23$ | $-1.25$ | $-0.10$ | $-2.09$ | - |
| $S_{PS}$ [2] (*p.s.u.*) | | 15.49 | 16.01 | 26.52 | 1.27 | $-0.90$ *** |
| NH$_4^+$ | (cmol$_c$ L$^{-1}$) | 20.48 | 20.47 | 32.62 | 2.16 | $-0.89$ *** |
| | (mg L$^{-1}$) | 3694 | 3693 | 5884 | 389 | - |
| Na$^+$ | (cmol$_c$ L$^{-1}$) | 7.78 | 7.34 | 27.92 | 0.52 | $-0.51$ *** |
| | (mg L$^{-1}$) | 1789 | 1688 | 6419 | 119 | - |
| K$^+$ | (cmol$_c$ L$^{-1}$) | 5.79 | 5.63 | 22.51 | 0.16 | $-0.68$ *** |
| | (mg L$^{-1}$) | 2263 | 2202 | 8802 | 64 | - |
| Mg$^{2+}$ | (cmol$_c$ L$^{-1}$) | 0.38 | 0.37 | 0.81 | 0.03 | $-0.47$ ** |
| | (mg L$^{-1}$) | 46.58 | 44.97 | 98.75 | 3.78 | - |
| Ca$^{2+}$ | (cmol$_c$ L$^{-1}$) | 1.17 | 0.72 | 10.26 | 0.20 | $-0.07$ |
| | (mg L$^{-1}$) | 234 | 144 | 2056 | 39 | - |
| Cl$^-$ | (cmol$_c$ L$^{-1}$) | 10.94 | 9.68 | 32.06 | 0.37 | $-0.71$ *** |
| | (mg L$^{-1}$) | 3879 | 3432 | 11,366 | 132 | - |
| NO$_3^-$ | (cmol$_c$ L$^{-1}$) | 0.03 | 0.00 | 0.08 | 0.00 | $-0.60$ *** |
| | (mg L$^{-1}$) | 17.90 | 0.00 | 47.22 | 0.00 | - |
| SO$_4^{2-}$ | (cmol$_c$ L$^{-1}$) | 0.12 | 0.04 | 1.14 | 0.00 | $-0.49$ ** |
| | (mg L$^{-1}$) | 58.97 | 17.33 | 549.30 | 0.00 | - |
| PO$_4^{3-}$ | (cmol$_c$ L$^{-1}$) | 0.88 | 0.79 | 4.07 | 0.00 | $-0.67$ *** |
| | (mg L$^{-1}$) | 279.07 | 250.35 | 1287.00 | 0.00 | - |

[1] Water potential, [2] Practical salinity, [3] Pearson correlation coefficient between the water potential and other parameters (** $p < 0.01$, *** $p < 0.001$).

The value of $\Psi$ of an anaerobic digester is mainly affected by ionic compounds (cations and anions), and as the concentration of ionic compounds increases, the $\Psi$ decreases [36,37]. In addition, organic acids, amino acids, polypeptides, and proteins with acidic dissociation functional groups also cause $\Psi$ to decrease in solution. However, among the ionic

compounds, multivalent ionic species such as $Ca^{2+}$, $Mg^{2+}$, $SO_4^{2-}$, and $PO_4^{3-}$ often exist as precipitated substances while forming complexes with each other, and these precipitated substances do not affect $\Psi$ [38–41]. In this study, the $NH_4^+$ concentration of the anaerobic digestate was 3694 mg/L, which was high compared to other cationic compounds, and the concentrations of $Na^+$ and $K^+$ were 1789 and 2263 mg $L^{-1}$, respectively. In this study, the types of ionic compounds that affect $\Psi$ were identified by measuring the ionic compounds present in soluble forms after removing TSS. Measurement of the concentration after the exclusion of ionic compounds that are present in the anaerobic digester but are inactive due to adsorption and precipitation revealed the concentration of $Cl^-$ to be the highest among those of the ionic compounds in the anaerobic digester. The physicochemical factors affecting the $\Psi$ of the anaerobic digester were evaluated by calculating the Pearson correlation coefficient, and $\Psi$ was correlated with the SCOD, TKN, $NH_4^+$-N, and alkalinity at a significance level of 99.9% ($p < 0.001$). The correlation between $\Psi$ and the solid content was significant at TS ($p < 0.05$) and VS ($p < 0.01$). $\Psi$ was correlated with $S_{PS}$, $NH_4^+$, $Na^+$, $K^+$, $Cl^-$, $NO_3^-$, and $PO_4^{3-}$ at a significance level of 99.9% ($p < 0.001$), and with $Mg^{2+}$ and $SO_4^{2-}$ at a significance level of 99.0% ($p < 0.01$).

The correlation between the $\Psi$ of the anaerobic digester and the $Ca^{2+}$ content was evaluated as very low. As for the correlation between $\Psi$ and ionic compounds, the divalent cations were reported to affect $\Psi$ [10,37], but in this study, the effect of polyvalent cations on $\Psi$ was evaluated as low. In this study, by evaluating the correlation between $\Psi$ and the soluble forms of the ionic compounds in the anaerobic digestate, the small effect on $\Psi$ of divalent cations such as $Ca^{2+}$ is considered to be the consequence of the relatively high degree of adsorption and precipitation of these cations. The SCOD is an indicator of soluble organic substances such as sugars and organic acids present in the anaerobic digester [42] and was significantly correlated with $\Psi$ of the anaerobic digester. TKN is an indicator of the organic (proteins and amino acids) and inorganic nitrogen ($NH_4^+$-N, $NH_3$-N) content of the anaerobic digester, and has a significant correlation with $\Psi$ [43]. In particular, $NH_4^+$-N is a weak acid (pKa = 9.245), and at pH lower than pH 9.245, the $NH_4^+$ ionic form is dominant, and above pH 9.245, the $NH_3$ gas form is dominant [44]. Therefore, TAN and $NH_4^+$-N of the anaerobic digester were strongly correlated ($p < 0.001$) with $\Psi$. The alkalinity is an indicator of the amount of $H^+$ buffer substances such as $HCO_3^-$, $CO_3^{2-}$, and $OH^-$, and had a strong correlation ($p < 0.001$) with $\Psi$ along with the pH ($p < 0.01$) [45].

### 3.2. Water Potential ($\Psi$) Inhibition Assay

In order to understand the effect of $\Psi$ on methane production in anaerobic digestion, the $\Psi$ of the anaerobic batch reactors was adjusted to $-0.40$ (Control), $-0.57$ (K1), $-0.89$ (K2), $-1.49$ (K3), $-2.85$ (K4), and $-3.91$ MPa (K5) and they were operated for 78 days. During this period, the production and consumption of TVFAs in the anaerobic reactor were delayed as $\Psi$ decreased (Table 6). The anaerobic digestion of organic matter proceeds via microbial chemical reactions in the stages of hydrolysis, acidogenesis, acetogenesis, and methanogenesis [46], and the results of this study showed that low $\Psi$ levels simultaneously had inhibitory effects on the methanogens as well as the microorganisms involved in acetogenesis. However, during the operation of the batch reactors, the $\Psi$ of the reactors was maintained as stable without significant fluctuations. These results showed that the effect of the variation in $\Psi$ according to the change in the concentration of the TVFAs was not significant during the operation of the anaerobic digester.

Figure 1 shows the cumulative methane yield curves in the $\Psi$ inhibition assay, and Table 5 lists the parameters and organic matter degradation rate ($VS_r$) derived by optimization of the modified Gompertz model. The value of $B_u$ was 0.264 $Nm^3$ $kg^{-1}$-$VS_{added}$ for the Control ($-0.40$ MPa). When $\Psi$ was adjusted, $B_u$ increased to 0.278 $Nm^3$ $kg^{-1}$-$VS_{added}$ at $-1.49$ MPa (K3), but decreased to 0.203 and 0.172 $Nm^3$ $kg^{-1}$-$VS_{added}$ at $-2.85$ MPa (K4) and $-3.91$ MPa (K5), respectively. The effect on methane production according to the decrease in $\Psi$ was clearly reflected in the maximum methane production rate ($R_m$) and lag growth phase time ($\lambda$); that is, $R_m$ for the Control ($-0.40$ MPa) was 8.11 mL day$^{-1}$ and

decreased to 1.70 mL day$^{-1}$ at K5 ($-3.91$ MPa), and λ was delayed to 35.96 and 25.34 days at K4 ($-2.85$ MPa) and K5 ($-3.91$ MPa), respectively (Table 7). Therefore, the level of Ψ that inhibits methane production in the anaerobic digestion of glucose was expected to be between $-1.49$ MPa (K$^+$ 2.22%) and $-2.85$ MPa (K$^+$ 3.07%) (Figure 2). In batch-type anaerobic digestion using cow manure, anaerobic digestion was reportedly inhibited by 50% at a K$^+$ concentration of 2.80% (*w*/*v*) [16], and a similar K$^+$ inhibitory concentration range was determined in this study. However, at very low Ψ levels of $-2.85$ MPa or less, the methane yield and methane production rate rapidly decreased, whereupon the activity of anaerobic microorganisms gradually recovered when sufficient acclimatization time elapsed.

**Table 6.** Variation in the concentration of TVFAs and water potential (Ψ) during the water potential inhibition assay.

| Parameters | Fermentation Time (Days) | Control | Treatments [1] | | | | |
|---|---|---|---|---|---|---|---|
| | | | K1 | K2 | K3 | K4 | K5 |
| TVFAs [2] (mg/L as acetate) | 0 | 105 (0.26) [4] | 104 (0.22) | 105 (0.25) | 105 (0.09) | 105 (0.21) | 105 (0.08) |
| | 12 | 1461 (0.09) | 1474 (3.26) | 1448 (7.71) | 1261 (2.44) | 1789 (10.08) | 932 (2.87) |
| | 45 | 109 (2.97) | 112 (0.81) | 107 (3.72) | 111 (1.04) | 1393 (5.99) | 1357 (5.95) |
| | 78 | 116 (0.33) | 123 (12.55) | 116 (0.56) | 115 (0.42) | 266 (1.33) | 399 (10.35) |
| Ψ [3] (MPa) | 0 | $-0.37$ (0.01) | $-0.55$ (0.05) | $-0.86$ (0.01) | $-1.46$ (0.01) | $-2.84$ (0.00) | $-3.94$ (0.01) |
| | 12 | $-0.39$ (0.01) | $-0.55$ (0.02) | $-0.88$ (0.00) | $-1.46$ (0.00) | $-2.84$ (0.03) | $-3.91$ (0.04) |
| | 45 | $-0.42$ (0.02) | $-0.62$ (0.01) | $-0.92$ (0.02) | $-1.56$ (0.01) | $-2.86$ (0.01) | $-3.92$ (0.03) |
| | 78 | $-0.43$ (0.040 | $-0.58$ (0.03) | $-0.93$ (0.02) | $-1.49$ (0.01) | $-2.86$ (0.00) | $-3.90$ (0.04) |

[1] Water potential was adjusted using KCl solution in the water potential inhibition assay, [2] Total volatile fatty acids, [3] Water potential, [4] Standard deviations.

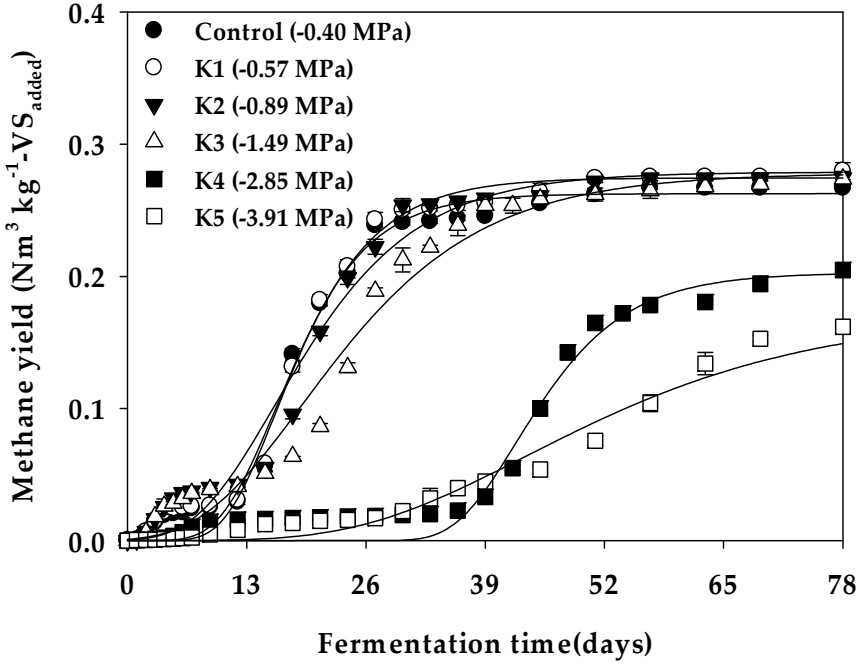

**Figure 1.** Cumulative methane yield curves in the water potential (Ψ) inhibition assay.

**Table 7.** Ultimate methane yield and the parameters calculated with the modified Gompertz model for the water potential (Ψ) inhibition assay.

| Parameters | Control | Treatments [1] | | | | |
|---|---|---|---|---|---|---|
| | | K1 | K2 | K3 | K4 | K5 |
| Ψ [2] (MPa) | −0.40 | −0.57 | −0.89 | −1.49 | −2.85 | −3.91 |
| VS$_r$ [3] (%) | 70.64 c | 73.63 b | 74.55 a | 74.54 a | 54.38 d | 46.21 e |
| Model parameters — B$_u$ [4] (Nm³ kg⁻¹-VS$_{added}$) | 0.264 c | 0.275 b | 0.278 a | 0.278 a | 0.203 d | 0.172 e |
| $p$ [5] (mL) | 127 c | 132 b | 134 a | 134 a | 98 d | 83 e |
| R$_m$ [6] (mL day⁻¹) | 8.11 a | 7.92 b | 5.84 c | 4.67 e | 5.48 d | 1.70 f |
| λ [7] (days) | 10.34 c | 10.16 c | 7.80 e | 8.65 d | 35.96 a | 25.34 b |
| R² | 0.990 | 0.990 | 0.984 | 0.983 | 0.982 | 0.985 |

[1] Water potential was adjusted using KCl solution in the water potential inhibition assay, [2] Water potential, [3] Ratio of anaerobic degradation (B$_u$/B$_{th}$ × 100; B$_{th}$ of D-glucose = 0.373 Nm³/kg-VS$_{added}$), [4] Methane potential, [5] Maximum methane production, [6] Maximum methane production rate, [7] Lag growth phase time. Mean with different letter differs significantly between treatment (DMRT; Duncan's multiple range test, $p < 0.05$).

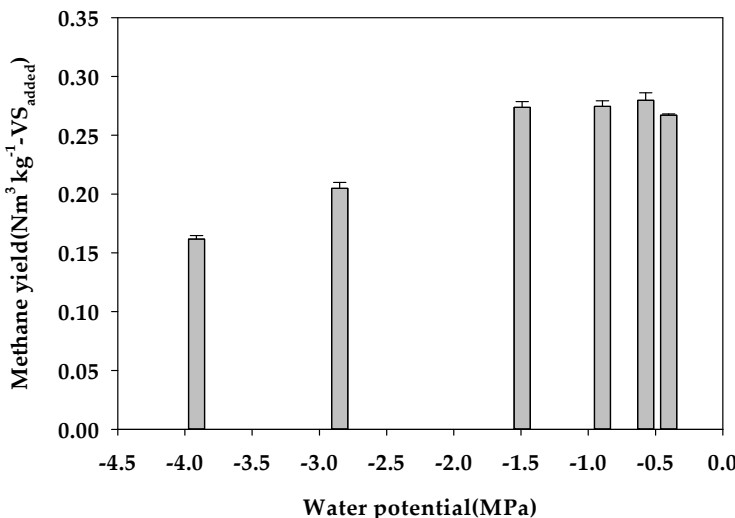

**Figure 2.** Effect of the water potential (Ψ) on the methane yield (vertical bars mean the standard deviations).

In general, methanogens are known to have survived even in high concentrations of salt [19,47], and it has been reported that, by accumulating osmotic substances, the intracellular osmotic potential is lowered and adapted to the low Ψ level of the surroundings [48–50]. In general, in a low Ψ environment induced by cations such as K⁺, plasmolysis occurs as water flows out of the cells. However, conversion of the environment of these cells to a hypotonic solution would enable the cells to recover to their normal condition [25,26], although the inhibition of Ψ by substances with a high molecular mass such as polyethylene glycol (PEG) results in cytorrhysis because of a strong decrease in intracellular pressure [26,51,52]. Therefore, the fact that gradual methane production proceeded even at the lowest Ψ level (−3.91 MPa) in this study indicates that moisture with a Ψ of −3.91 MPa did not have a fatal effect such as the collapse of the cell wall of anaerobic microorganisms. This was evaluated as the Ψ level at which cell activity could be restored by acclimatization. Figure 3 shows the degree of inhibition of methane production according to the decrease in Ψ and the increase in the K⁺ concentration, respectively. Analysis of the Ψ inhibition level using linear regression in the Ψ range in which the methane yield was inhibited owing to the decrease in Ψ enabled us to predict that the methane yield was inhibited from about −1.65 MPa. At this time, the K⁺ concentration at which the methane yield was inhibited was found to be 11.69 g L⁻¹. Several researchers conducted anaerobic digestion toxicity studies by K⁺ during anaerobic digestion (Table 8) and reported the concentration

of $K^+$ capable of inhibiting anaerobic digestion according to the feeding materials and operating temperatures in various reactors. In this study, the concentration of $K^+$ at which anaerobic digestion is inhibited (11.69 g $L^{-1}$) was within the inhibition range reported by previous researchers. Generally, the inhibition effect of $K^+$ is rarely referenced in the literature. Low concentrations of potassium (less than 400 mg $L^{-1}$) were reported to enhance the performance of both the thermophilic and mesophilic ranges, whereas there was an inhibitory effect in the thermophilic temperature range at higher $K^+$ concentrations. However, the concentration of $K^+$ at which anaerobic digestion is inhibited varies from 3000 to 28,000 mg $L^{-1}$ depending on the type of feeding material and the temperature [5,11,14–16], and the mechanism of metabolic inhibition of anaerobic microorganisms by $K^+$ has not been reported in detail. This suggests that the inhibitory effect on anaerobic digestion by K+ may be an indirect inhibitory effect due to the decrease in Ψ that accompanies high concentrations of $K^+$ rather than a direct toxic effect of $K^+$.

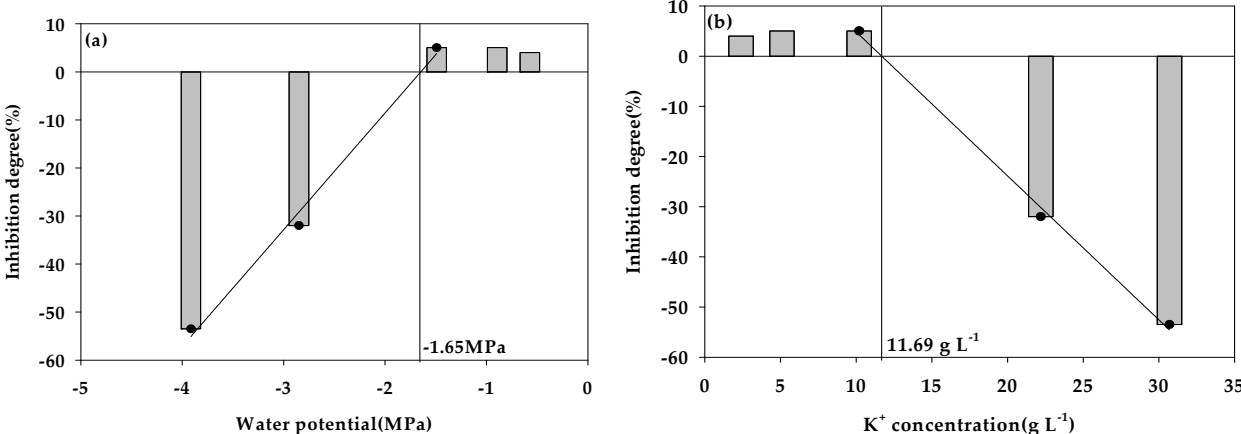

**Figure 3.** Effects of the water potential (Ψ) (**a**) and $K^+$ concentration (**b**) on the degree of inhibition of methane production from food wastewater.

**Table 8.** Stimulatory and inhibitory concentrations of $K^+$ on anaerobic digestion.

| Reactor Type | Temperature | Substrate | Concentrations (mg $L^{-1}$) | | Reference |
|---|---|---|---|---|---|
| | | | Stimulatory | Inhibitory | |
| UASB [1] | 35 °C | IW [3], SS [4] | 200~400 | 12,000 | [14] |
| CSTR [2] | 50 °C | PS [5] | - | 3000 | [18] |
| Batch | 55 °C | CM [6] | - | 28,000 (50% inhibition) | [16] |
| Batch | 35 °C | MA [7] | - | 13,000 (52~53% inhibition) | [15] |
| Batch | 54 °C | SS | - | 8000 (50% inhibition) | [11] |

[1] Upflow anaerobic sludge blanket, [2] Continuous stirred tank reactor, [3] Industrial waste, [4] Sewage sludge, [5] Pig slurry, [6] Cow manure, [7] Macroalgae.

### 3.3. Microbial Community

The effect of Ψ on changes in the characteristics of a community of microorganisms in anaerobic digestion was investigated by performing the Ψ inhibition assay in the Ψ range of −0.40 to −3.91 MPa. Then, after the batch reactor had been in operation for 78 days, the characteristics of a community of microorganisms were analyzed using the NGS method. During anaerobic digestion, microbial methane is mainly produced by the methanogenic archaea, which could produce methane primarily from $CO_2/H_2$, methyl groups, or acetate under anaerobic conditions [53]. Methanogenic archaea are slow-growing microorganisms, and methane is produced by keeping the syntrophic mechanism with variable microorganisms concerned in the acetogenesis [54]. Methanogenic archaea are assumed to belong to the *Euryarchaeota* phylum. In the *Euryarchaeota* phylum, the methanogenic archaea are

classified into eight orders, namely *Methanobacteriales*, *Methanococcales*, *Methanomicrobiales*, *Methanosarcinales*, *Methanopyrales*, *Methanocellales*, *Methanomassiliicoccales*, and *Methanona-tronarchaeales* [54,55]. In this study, the four orders of *Methanobacteriales*, *Methanomicrobiales*, *Methanomassiliicoccales*, and *Methanosarcinales* constituted the community of methanogens (Figure 4), and the community of methanogens accounted for 73.47 to 80.14% of the total microbial community.

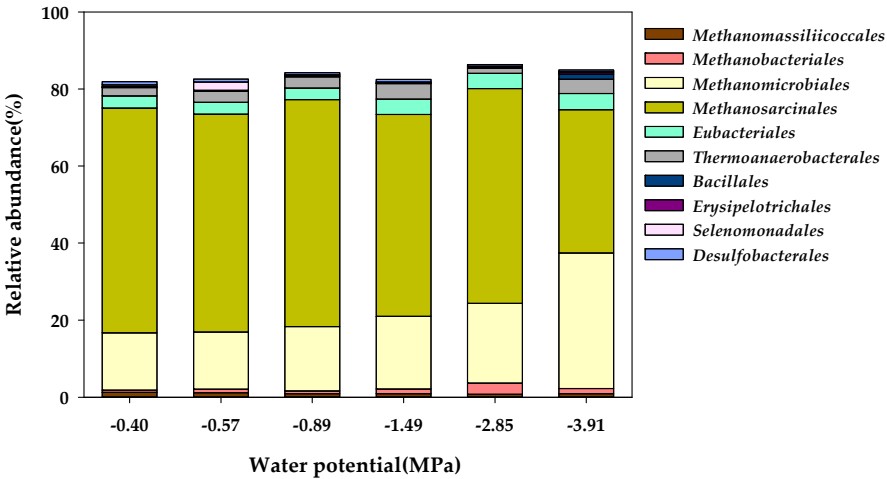

**Figure 4.** Order-level taxonomic classification of the microbial community in the water potential (Ψ) inhibition assay.

In particular, *Methanosarcinales* had the largest distribution in the methanogen community, with 58.34 in the Control (−0.40 MPa), 56.56 in K1 (−0.57 MPa), 58.88 in K2 (−0.89 MPa), 52.38 in K3 (−1.49 MPa), 55.80 in K4 (−2.85 MPa), and 37.19% K5 (−3.91 MPa), with a rapid decrease in the cluster distribution at the Ψ level of K5 (−3.91 MPa). Although the distribution of *Methanomicrobiales* was 14.85% in the Control (−0.40 MPa), the community distribution increased from 14.84% in K1 (−0.57 MPa) to 35.19% in K5 (−3.91 MPa) as Ψ decreased. The proportion of *Methanomassiliicoccales* in the methanogen community was as low as 1.27% in the Control (−0.40 MPa) and decreased from 1.21% in K1 (−0.57 MPa) to 0.92% in K5 (−3.91 MPa) according to the decrease in Ψ. In addition, the proportion of *Methanobacteriales* among methanogens was as low as 0.59% in the Control (−0.40 MPa). However, it showed community characteristics that increased from 0.86% in K1 (−0.57 MPa) to 1.33% in K5 (−3.91 MPa) according to the decrease in Ψ. The methanogenesis process consists of four main pathways: $CO_2$ reduction, acetoclastic, methylotrophic, and methyl reduction pathways [55]. The *Methanosarcinales* order is capable of undergoing methanogenesis via all four methanogenic pathways. Generally, the *Methanomicrobiales* order can reduce $CO_2$ to methane with electrons derived from the oxidation of $H_2$ via the $CO_2$ reduction pathway [56]. Contrary to this, the species in the *Methanobacteriales* and *Methanomassiliicoccales* orders proceed along a methyl reduction pathway by reducing methyl compounds to methane with electrons derived from $H_2$ [55,57]. In the genus-level taxonomic classification of the microbial community, *Methanosarcina* and *Methanothrix* were found to be methanogens belonging to the *Methanosarcinales* order, and the community proportion of *Methanosarcina*, reported as an acetoclastic methanogen [58], decreased significantly to 36.76% in K5 (−3.91 MPa) in comparison to the 58.15% in the Control (−0.40 MPa) (Figure 5). However, in the *Methanomicrobiales* order, *Methanoculleus*, which has been reported as a hydrogenotrophic methanogen [58], formed the majority of the colony, and it significantly increased to 35.16% in K5 (−3.91 MPa) compared to 14.85% in the Control (−0.40 MPa).

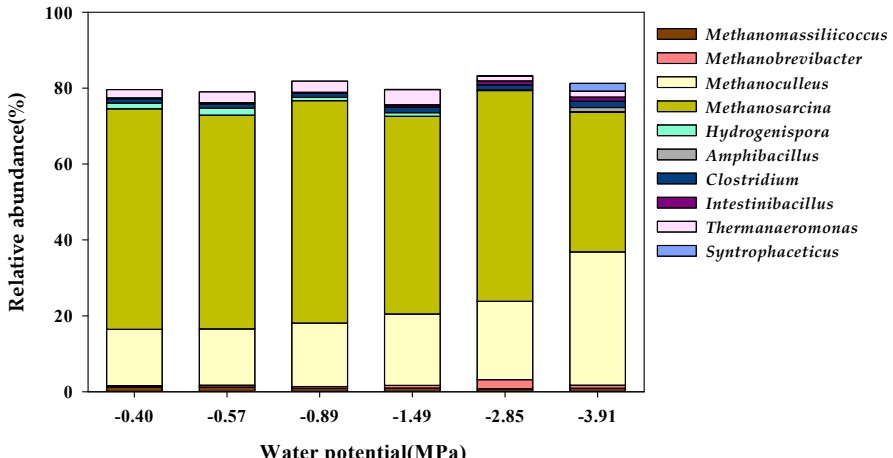

**Figure 5.** Genus-level taxonomic classification of the microbial community in the water potential ($\Psi$) inhibition assay.

Therefore, in this study, at a low $\Psi$ level of $-3.91$ MPa, the population of acetoclastic methanogens (*Methanosarcina*) decreased and the population of hydrogenotrophic methanogens (*Methanoculleus*) increased, and these results suggested that hydrogenotrophic methanogens are advantageous for acclimation under low $\Psi$ conditions. In general, hydrogenotrophic methanogens are syntrophic partners that are associated with microorganisms in the acetogenesis. This led to the speculation that changes in the community characteristics of the methanogen according to the decrease in $\Psi$ are related to the decrease in the activity of microorganisms at the stages of acidogenesis and acetogenesis. In this study, the production of TVFAs was increasingly delayed during the operation of the anaerobic reactor as the $\Psi$ level decreased (Table 6), and these results showed that the microorganisms in the acidogenesis and acetogenesis stages were also inhibited by the low $\Psi$. In particular, syntrophism occurs as a result of indirect interspecies electron transfer (IIET), in which hydrogen or formate, a product of acetogenesis, acts as an electron carrier between methanogens [59]. This means that, even though the adaptation mechanism of hydrogenotrophic methanogens to low $\Psi$ is important to increase the community of hydrogenotrophic methanogens at a low $\Psi$, the adaptation mechanism of acetogenic microorganisms related to IIET should also be considered. Microorganisms belonging to the *firmiutes* phylum are acidogenic and acetogenic and are involved in the production of VFAs and acetate [60,61]. In particular, the *Eubacteriales* and *thermoanaerobacterales* orders are involved in acetogenesis, producing acetate, $H_2$, and $CO_2$. In this study, *Clostridium* belonging to the *Eubacteriales* order and *Syntrophaceticus* belonging to the *thermoanaerobacterales* order maintained or increased their community distribution at a low $\Psi$ (Figure 5). In addition, *Clostridium* maintained a range of 1.05 to 1.64% in the $\Psi$ range from $-0.40$ MPa to $-3.91$ MPa, and *Syntrophaceticus* had a population of 0.00% in the Control ($-0.40$ MPa) but it increased to 2.09% in K5 ($-3.91$ MPa).

## 4. Conclusions

This study was carried out to understand the effect of $\Psi$ on anaerobic methane production and a microbial consortium. The $\Psi$ of the investigated anaerobic digester (n = 20) ranged from $-0.10$ to $-2.09$ MPa with a mean value of $-1.23$ MPa, and the $\Psi$ of the anaerobic digester was significantly correlated with the SCOD, TKN, $NH_4^+$-N, alkalinity, $S_{PS}$, $NH_4^+$, $Na^+$, $K^+$, $Cl^-$, $NO_3^-$, and $PO_4^{3-}$ ($p < 0.001$). In the $\Psi$ inhibition assay, the $R_m$ of the Control ($-0.40$ MPa) was 8.11 mL day$^{-1}$ and decreased to 1.70 mL day$^{-1}$ at $-3.91$ MPa (K5), and the $\lambda$ was delayed to 35.96 and 25.34 days at $-2.85$ MPa (K4) and $-3.91$ MPa (K5), respectively. The $B_u$ was 0.264 Nm$^3$ kg$^{-1}$-VS$_{added}$ for the Control, and $B_u$ increased to 0.278 Nm$^3$ kg$^{-1}$-VS$_{added}$ at $-1.49$ MPa (K3) but decreased to 0.203 and 0.172 Nm$^3$ kg$^{-1}$-VS$_{added}$ at $-2.85$ MPa (K4) and $-3.91$ MPa (K5), respectively. Therefore, the methane

yield was inhibited due to the decrease in the $\Psi$, with inhibition of the methane yield predicted to occur from about $-1.65$ MPa, which corresponded to the $K^+$ concentration of 11.69 g $L^{-1}$. In the genus-level taxonomic classification of the microbial community, the relative abundance of *Methanosarcina* (acetoclastic methanogens) decreased significantly to 36.76% (58.15% for the Control) at the lowest $\Psi$ level; however, the relative abundance of *Methanoculleus* (hydrogenotrophic methanogens) significantly increased to 35.16% (14.85% for the Control) at this $\Psi$ level. In this study, not only methane-producing bacteria but also acid- and acetic acid-producing microorganisms were expected to be inhibited by low $\Psi$. Existing studies on salt inhibition in an anaerobic digester mainly focused on different types of cations ($Na^+$, $K^+$, $Ca^{2+}$, $Mg^{2+}$) and examined the phenomenon of inhibition of the anaerobic digestion efficiency by individual cations. $\Psi$ is a colligative property that depends on the ratio of the number of solute particles to the number of solvent particles in a solution. In addition, $\Psi$ is not only affected by the salt content but also by various micro- and macromolecules with TVFAs, amino acids, and proteins that contain electronic functional groups present in the anaerobic digester. However, few studies on the effect of $\Psi$ on anaerobic digestion efficiency have been reported thus far, and future research on the inhibition of anaerobic digestion efficiency by cations would require in-depth research in relation to $\Psi$, which represents the colligative properties of the solution. These studies are anticipated to make it possible to use $\Psi$ as a new operating factor for salt management in the operational management of an anaerobic digester.

**Author Contributions:** Conceptualization, Y.-M.Y. and J.Y.; methodology, J.Y.; software, J.-H.L. and E.S.; validation, C.-G.K., E.S. and Y.-M.Y.; formal analysis, J.Y.; investigation, J.Y.; resources, Y.-M.Y.; data curation, Y.-M.Y.; writing—original draft preparation, J.Y. and C.-G.K.; writing—review and editing, Y.-M.Y.; visualization, J.-H.L.; supervision, Y.-M.Y.; project administration, Y.-M.Y.; funding acquisition, Y.-M.Y. All authors have read and agreed to the published version of the manuscript.

**Funding:** This study was funded by MOLIT and KAIA's development of combined plant compact technology (Project No. RS-2020-KA157945).

**Institutional Review Board Statement:** Not applicable.

**Informed Consent Statement:** Not applicable.

**Acknowledgments:** This study was supported through MOLIT and KAIA's development of combined plant compact technology (Project No. RS-2020-KA157945).

**Conflicts of Interest:** The authors declare no conflict of interest. The funders had no role in the design of the study; in the collection, analyses, or interpretation of data; in the writing of the manuscript, or in the decision to publish the results.

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
