# Peer review of "Effects of Water Potential on Anaerobic Methane Production and a Microbial Consortium"

_fermentation, doi:10.3390/fermentation9030244_

Round 1

Reviewer 1 Report

Comments to the Authors:

In their work the authors evaluated the effect of water potential on methane production in anaerobic digestion. They analyzed an adequate number of digesters with different conditions and studying the physico-chemical properties or microbial community in each one and summarized the results in an adequate number of graphs and tables. In my opinion this work can help to improve the knowledge of methane production under anaerobic digestion for designing more suitable processes. Nevertheless, some questions are necessary clarify for improving the quality of this work. I indicate below

1.-In line 135, the authors wrote “…methane production data to be optimized using these equations”. Please, replace by “…methane production data to be optimized using the follows equation”. The authors only indicate one equation, so the singular form is required.

2.-Replace digestates by digesters

3.-In line 215 and section: 3.1. Correlation factors with Ψ, the authors wrote “20 anaerobic digesters operating in Korea” If they want to indicate the place, please providing a GPS information of digesters or most concrete area, because Korea did not indicate any information.

4.-In Table 4, what is the meaning of subscript “cr” in the abbreviations “TCOD” and “SCOD”?

5.- The authors indicated in table 4 that the pH range on anaerobic digesters was (6.8-7.8). Was this pH range influential in the results obtained or the pH is not a variable to take into account in this study? Give an explanation and emphasizing this point in the manuscript

6.- In figure 3, what is the meaning of “kg-VSadded” in the ordinate axis legend “Methane yield(Nm3/kg-VSadded)?

Author Response

We have revised the manuscript as reviewer’s comments.
Thank you so much.

Comment 1. In line 135, the authors wrote “methane production data to be optimized using these equations”. Please, replace by “methane production data to be optimized using the follows equation”. The authors only indicate one equation, so the singular form is required.

Response: Thank you. As pointed out, the text has been corrected.

Comment 2. Replace digestates by digesters

Response: "digesters" is the concept of the space or process in which anaerobic digestion takes place, and "digestates" is the concept of the result or substance of anaerobic digestion. Therefore, I think it is appropriate to use "digestates" in this article.

Comment 3 In line 215 and section: 3.1. Correlation factors with Ψ, the authors wrote “20 anaerobic digesters operating in Korea” If they want to indicate the place, please providing a GPS information of digesters or most concrete area, because Korea did not indicate any information.

Response: Thank you. In this study, 20 anaerobic digesters operating in Korea were investigated. As pointed out, the location information of 20 anaerobic digestion facilities has been added to Table 1.

Comment 4. In Table 4, what is the meaning of subscript “Cr” in the abbreviations “TCOD” and “SCOD”?

Response: Subscript “Cr” means chrome (Cr). Internationally, COD is measured using the oxidizing power of K2Cr2O7(Potassium dichromate (VI)), whereas in Korea, it is measured using the oxidizing power of KMnO4(Potassium manganate (VII)). However, since the COD value is measured differently due to the difference between the oxidizing power of Cr and that of Mn, the oxidizing agent used is expressed as CODMn or CODCr in the COD measurement value. TCOD is the COD value measured in the total solid, and SCOD is the COD value measured in the filtrable solid. In general, SCOD is measured to determine the content of soluble organic matter.

Comment 5. The authors indicated in table 4 that the pH range on anaerobic digesters was (6.8-7.8). Was this pH range influential in the results obtained or the pH is not a variable to take into account in this study? Give an explanation and emphasizing this point in the manuscript

Response: Thank you. pH is the most commonly used anaerobic digester operating factor. Optimal pH conditions for methane production are around 6.8 8.3. In this study, there was no major problem in the operating conditions of the 20 anaerobic digesters investigated, and the pH results were mentioned to indicate that they were operating under appropriate operating conditions.

Comment 6. In figure 3, what is the meaning of “kg-VSadded” in the ordinate axis legend “Methane yield (Nm3/kg-VSadded)?

Response: “kg-VSadded” is a unit representing methane yield. It means the production of CH4 per organic matter (VS) input (“added”) to the serum bottle for BMP assay. Here, “N” means “normal state (0, 1atm)”, and “kg-VSadded” is the amount of organic matter added. Another expression is “Nm3/kg-VSremoved”, which means the amount of CH4 produced per organic matter decomposed(“removed”) during anaerobic digestion.

Young-Man Yoon

Reviewer 2 Report

Authors did great job. This paper should be accepted in present form.

Author Response

We have revised the manuscript as per the reviewer’ comments.

Thank you so much.

Comment 1. Authors did great job. This paper should be accepted in present form.

Response: Thank you so much. Water potential is an important optimizing factor of an anaerobic digester. Through further study, we will try to develop a plan to utilize water potential as an anaerobic digester operating factor.

Young-Man Yoon

Reviewer 3 Report

Fermentation 2243996

Effects of Water Potential on Anaerobic Methane Production and a Microbial Consortium

Overall

Yeo et al. present the analysis of two linked experiments. In the first, samples were collected from 20 operating anaerobic digesters and analysed; in the second, water potential was manipulated across a range of treatments to test the effects on methane production and the factors that may affect the methanogenic organisms fermenting the materials.

The large number of factors analysed across two different experiments, many of which have names or abbreviations that are not in wide use, makes the presentation of key results difficult. More care needs to be taken with the overall structure of this manuscript, with greater use of figures and tables, and greater explanation of main patterns. The overall structure of this manuscript is unusual, with a combined Results and Discussion section that is overwhelmingly a presentation of results, with little discussion attached, followed by a separate Conclusions that is written as an introduction to the topic of water potential and methanogenesis for a naïve audience, without any clear statements of conclusion regarding the preceeding manuscript.

I think there are interesting and useful results in this study, and the stated motivation is good – water potential as a broad collective variable may be very important when optimising methane production, and the nuances of microbial acclimatization are worthy of further investigation.

Abstract

The abstract includes many undefined terms, such as K5 (one of the treatments in one of the experiments), SCODcr (oxygen demand of a component of some solutions), and the greek letter lambda (defined under Equation 5 as the lag growth phase time in days, LN139). The abstract would be more informative if fewer details were presented and more terms were defined, or common-use terms were used instead. For example, the sentence on LN14-16, starting with “The maximum methane production rate” could be simplified and made more comprehensible to a wider audience if the “(Rm)”, “K5”, “K4”, and “[lambda]” were removed, and words such as “some treatments” or “lag growth phase time” were used.

Results and Discussion

Table 4. What does the column for Pearson’s r represent? What is being correlated against each parameter? On LN241-243, it seems to be water potential that is correlated with some parameters, but this is not well described – am I correct in this?

LN263-272 might usefully be replaced as a table rather than main text. This text compares TVFA contents of the anaerobic bath reactors in control plus five treatments over 78 days, with measurements of all treatments at fixed, consistent times. This seems like an ideal type of data to display as a table that allows the reader to easily see differences and similarities across treatments, rather than repeatedly searching the paragraph.

Or, given that Figure 1 appears to display the same data, this paragraph could be shortened considerably. There is no need to state exact values in main text when a figure is also present that allows rapid and easy comparison.

Conclusions

The conclusion reads like a literature review or introduction to the topic of water potential in an academic textbook. Where is the summation of the key findings of this study? Where are the specific recommendations to further this line of research? Some of the text here may be useful in the Introduction, but most of it should be deleted, and replaced with a concise summary of the most important results of the study.

Author Response

We have revised the manuscript as the reviewer’ comments.

Thank you so much.

Comment 1. The abstract includes many undefined terms, such as K5 (one of the treatments in one of the experiments), SCODCr (oxygen demand of a component of some solutions), and the greek letter lambda (defined under Equation 5 as the lag growth phase time in days, LN139). The abstract would be more informative if fewer details were presented and more terms were defined, or common-use terms were used instead. For example, the sentence on LN14-16, starting with “The maximum methane production rate” could be simplified and made more comprehensible to a wider audience if the “(Rm)”, “K5”, “K4”, and “[lambda]” were removed, and words such as “some treatments” or “lag growth phase time” were used.

Response: Thank you. In order to enhance the readability of abstracts, Abstracts were revised substantial words were used rather than abbreviations.

Comment 2. Table 4. What does the column for Pearson’s r represent? What is being correlated against each parameter? On LN241-243, it seems to be water potential that is correlated with some parameters, but this is not well described am I correct in this?

Response: Thank you. Pearson’s r means the Pearson correlation coefficient between the water potential and other parameters. Therefore, the column for Pearson's r means the Pearson correlation coefficient with each corresponding parameter. To help understand Table 4, an explanation of Pearson's r has been added to the annotations.

Comment 3. LN263-272 might usefully be replaced as a table rather than main text. This text compares TVFA contents of the anaerobic bath reactors in control plus five treatments over 78 days, with measurements of all treatments at fixed, consistent times. This seems like an ideal type of data to display as a table that allows the reader to easily see differences and similarities across treatments, rather than repeatedly searching the paragraph. Or, given that Figure 1 appears to display the same data, this paragraph could be shortened considerably. There is no need to state exact values in main text when a figure is also present that allows rapid and easy comparison.

Response: Thank you. As pointed out, Figure 1 was replaced with a Table (Table 6) to accurately understand the difference between treatments. Also, the related text has been revised.

Comment 4. The conclusion reads like a literature review or introduction to the topic of water potential in an academic textbook. Where is the summation of the key findings of this study? Where are the specific recommendations to further this line of research? Some of the text here may be useful in the Introduction, but most of it should be deleted, and replaced with a concise summary of the most important results of the study

Response: Thank you. The conclusion was revised with a concise summary of the most important results of the study.

Young-Man Yoon
